# Distinct Prostate Cancer Survival Outcomes in Firefighters: A Population-Based Study

**DOI:** 10.3390/cancers16071305

**Published:** 2024-03-27

**Authors:** Paulo S. Pinheiro, Tulay Koru-Sengul, Wei Zhao, Diana R. Hernandez, Monique N. Hernandez, Erin N. Kobetz, Alberto J. Caban-Martinez, David J. Lee

**Affiliations:** 1Department of Public Health Sciences, University of Miami Miller School of Medicine, Miami, FL 33136, USA; tsengul@med.miami.edu (T.K.-S.); ekobetz@med.miami.edu (E.N.K.); acaban@med.miami.edu (A.J.C.-M.); dlee@med.miami.edu (D.J.L.); 2Sylvester Comprehensive Cancer Center, University of Miami Miller School of Medicine, Miami, FL 33136, USA; wzhao2@med.miami.edu (W.Z.); drh118@miami.edu (D.R.H.); 3Florida Cancer Data System, Sylvester Comprehensive Cancer Center, University of Miami Miller School of Medicine, Miami, FL 33136, USA; mhernandez5@med.miami.edu; 4Department of Medicine, University of Miami Miller School of Medicine, Miami, FL 33136, USA

**Keywords:** prostate cancer, survival, firefighters, Florida, occupational exposure

## Abstract

**Simple Summary:**

Prostate cancer survival among US firefighters, who undergo regular medical check-ups and have unique exposures, has not been well-studied. Using population-based Florida cancer registry data, this retrospective study compared 1058 firefighter prostate cancer cases to patients with prostate cancer in the general population, finding significantly better five-year survival (96.1% vs. 94.2%) in firefighters. Firefighters were diagnosed earlier, had more localized cancers, and underwent surgery more often. However, older firefighters with regional or distant-stage cancer faced a higher risk of death. Enhanced survival in firefighters may be due to increased PSA testing, but further research is needed to understand factors influencing aggressive disease development and the impact of surgical treatments on their quality of life.

**Abstract:**

**Introduction:** Survival outcomes for prostate cancer among specific occupational groups prone to regular medical check-ups vis-à-vis the general population have been understudied. For firefighters, a demographic subject to rigorous medical evaluations, possessing above-average medical expertise, and exposed to specific carcinogens of interest, prostate cancer survival in the US has never been studied. **Methods:** We conducted a retrospective study, utilizing data from the Florida Cancer Data System spanning 2004 to 2014, coupled with firefighter certification records from the Florida State Fire Marshal’s Office. Our study cohort consisted of 1058 prostate cancer cases among firefighters as well as prostate cases for the Florida general population (*n* = 150,623). We compared cause-specific survival between the two using Cox regression models adjusted for demographics and clinical characteristics, including PSA levels, Gleason scores, and treatment modalities. **Results:** Firefighters demonstrated a higher five-year cause-specific survival rate (96.1%, 95% CI: 94.7–97.1%) than the general population (94.2%, 95%CI: 94.1–94.3%). Overall, firefighters’ diagnoses were established at younger ages (median age 63 vs. 67 in the general population), exhibited a higher proportion of localized stage cancers (84.7% vs. 81.1%), and had a greater utilization of surgery (46.4% vs. 37.6%), a treatment modality with a high success rate but potential side effects. In multivariable analysis, firefighters displayed a survival advantage for localized stage (adjusted hazard ratio [aHR] = 0.53; 95%CI: 0.34–0.82). However, for regional or distant stages, firefighters aged 65 and above exhibited a higher risk of death (aHR = 1.84; 95% CI: 1.18–2.86) than the general population. **Conclusion:** Firefighters experience enhanced prostate cancer survival, primarily in cases diagnosed at localized stages, likely due to increased PSA testing. Nonetheless, for regional or distant stage, survival among older firefighters’ lags behind that of the general population. Further investigations are warranted to unravel factors influencing the development of aggressive disease beyond PSA and Gleason scores in this population, as well as to assess the impact of a higher rate of surgical treatment on firefighters’ quality of life.

## 1. Introduction

Prostate cancer stands as a predominant concern among males in the United States (US). Prostate cancer ranks as the most common cancer and the second leading cause of death among men, and is responsible for over 10% of all cancer-related fatalities [1,2]. In the year 2023, an estimated 288,300 new cases of prostate cancer were anticipated to emerge in the US, accompanied by approximately 34,700 prostate cancer-related deaths [2]. During the period of 2015–2019, the age-adjusted prostate cancer incidence and death rate in the US were recorded at 109.9 and 18.8 per 100,000, respectively [3,4]. This significant divergence in incidence and mortality rates is a direct consequence of the remarkable survival rates associated with this disease. Notably, prostate cancer boasts a 5-year relative survival rate of 97% for all stages combined in the US, surpassing the survival rates of any other major cancers [5]. However, this relative survival rate varies significantly based on tumor stage, plummeting to as low as 32% for individuals diagnosed at an advanced, distant stage; fortunately, more than 80% of all cases are detected at a localized stage [5].

Prostate cancer is associated with older age, family history, and a higher prevalence among individuals of Black race [6]. Despite extensive research, the quest for other specific and modifiable risk factors has remained elusive. Importantly, the incidence of prostate cancer is closely tied to the prevalence of prostatic-specific antigen (PSA) testing as a common screening tool within a given population. This complexity adds layers to the study of prostate cancer, introducing associated biases such as lead time and overdiagnosis, which can significantly impact our understanding of both incidence and survival.

Firefighters, as a distinct occupational group, face frequent exposure to a multitude of toxic substances through the inhalation of particulate matter and gases, as well as direct bodily contact with various hazardous agents. The spectrum of cancer-causing toxins encountered on the job is extensive and includes carbon monoxide, benzene, sulfur dioxide, hydrogen cyanide, acrolein, aldehydes, hydrogen chloride, nitrogen dioxide, chlorinated hydrocarbons, trichloroethylene, toluene, dichlorofluoromethane, and soot [7,8]. In 2022, the World Health Organization’s International Agency for Research on Cancer (IARC) reclassified firefighting as a Group 1 carcinogenic profession [9,10]. However, the direct link between firefighting and prostate cancer remains supported only by limited evidence. Firefighters are subject to specific job health requirements and mandatory occupational health check-ups, which could lead to heightened diagnostic intensity for prostate cancer via the PSA test. Unsurprisingly, there is some evidence supporting a direct link between firefighting and prostate cancer incidence [11,12,13], but there is no such evidence for mortality. Nevertheless, given the numerous exposures inherent to firefighting, it is plausible that prostate cancer patterns and survival outcomes may differ between firefighters and the general population [9,11,12,14,15]. Furthermore, the lasting impact of these exposures, which may persist after retirement, could potentially determine different patterns of prostate cancer in this occupational group. Elsewhere a Norwegian study has indicated that prostate cancer survival among firefighters surpasses that of the general population, but to our knowledge, no study has ever been carried out in the US [16].

This study aims to address this knowledge gap by comparing prostate cancer characteristics and survival outcomes among firefighters to non-firefighters. The current research has two primary objectives: firstly, to compare specific patterns of prostate cancer among firefighters and non-firefighters, including pertinent prognostic factors such as PSA levels, Gleason scores, treatment modalities, and the stage at diagnosis for the period 2004–2014; secondly, to assess prostate cancer-specific survival among firefighters in Florida (both career and volunteer) comparing them with the general population in Florida. This study represents the first population-based analysis of the epidemiology and survival of prostate cancer among firefighters in the state of Florida.

## 2. Materials and Methods

### 2.1. Study Design

This retrospective observational cohort study is part of the Firefighters Cancer Initiative of the State of Florida [17,18,19,20].

### 2.2. Data Sources

This study uses incident cancer records from the Florida Cancer Data System (FCDS) as the analytical cohort. The data on firefighters were primarily obtained from the FCDS data linked to the firefighter certification records from the Florida State Fire Marshal’s Office (FMO) (1972–2012). Moreover, FCDS, as a member of the National Program of Cancer Registries (NPCR), collects standardized information about all cancer patients [19,21], including text indicating usual occupation (“type of job patient engaged in for the greatest number of working years”) and text indicating usual industry (“type of business or industry where patient worked in his or her usual occupation”). FCDS then applies the US census occupation and industry coding system to translate reported text fields into coded variables after consolidation of information with LexisNexis@, a national dataset of legal, government, business, and high-tech information for identification of missing linkage variables (e.g., date of birth, social security number). Firefighters who were not identified via data linkage but with US census-derived occupation codes in the cancer registry record as 3740, 3720, and 3750 were included as firefighters. In our study, the binary occupation variable (firefighter or non-firefighter) is considered as the predictor variable in the regression models. The design and methods of the original data linkage have been published previously [17].

### 2.3. Study Population

From both the FCDS (general population) and the FCDS-linked dataset (firefighters), all male patients in Florida with a single primary or a first of multiple primaries of prostate cancer (International Classification of Diseases for Oncology, third edition (ICD-O-3) primary site codes C61.9 and sequence numbers 0 and 1 from cancer registry) diagnosed between 2004 and 2014 were included. Vital status and follow-up duration were verified using records from the Florida Office of Vital Statistics and the National Death Index (2004–2019). To accrue at least 5 years of follow-up for all participants, the follow-up time for determining death due to cancer was set as 2019. Therefore, the timeframe for analysis includes 2004–2014 for cancer diagnosis and 2004–2019 for survival follow-up.

### 2.4. Prostate Cancer Specific Death as a Primary Clinical Outcome

Survival from prostate cancer was studied using the elapsed time in days from the date of cancer diagnosis to the earliest of the two dates: date of death or 31 December 2019, for alive patients. Causes of death not related to prostate cancer were considered as censored observations. Surveillance, Epidemiology, and End Results (SEER) standards based on the sequence number of prostate cancer were used to identify cause-specific deaths [22,23]. Cause-specific survival time in days was converted into months and years for easy interpretation.

### 2.5. Covariables Used for Adjustment

Variables used as additional covariables included year of cancer diagnosis, age at cancer diagnosis (years), race, ethnicity, health insurance, neighborhood level socioeconomic status (SES), cigarette use, SEER tumor stage (localized, regional, and distant), and treatment received, namely surgery and/or radiation therapy. In addition, important prognostic characteristics for prostate cancer at diagnosis such as PSA level and Gleason score were included.

### 2.6. Statistical Data Analysis Methods

We summarize the demographics and clinical characteristics of patients with prostate cancer stratified by occupation (firefighters and non-firefighters) in Table 1. Differences in occupation groups were assessed using the Chi-square test for independence. Survival analysis was conducted using the Kaplan–Meier method to determine survival proportions at 1, 3, 5, and 10 years (Table 2). To assess differences among occupational groups, log-rank tests were employed as shown in Figure 1. For the analysis of prostate cancer-specific mortality, we applied both univariable and multivariable Cox proportional hazard regression models. Occupation was the primary variable of interest in these models (Table 3 and Table 4). The multivariable models included the additional covariates listed above.

The identification of numerous non-aggressive localized cancers through widespread PSA testing, potentially occurring at varying rates between firefighters and non-firefighters, prompted us to further address this potential source of bias. Consequently, we performed stratified analyses based on the cancer stage at diagnosis, distinguishing between localized and regional/distant stages. Additionally, we stratified our analysis using the conventional retirement age threshold of 65 years (<65 and ≥65) to explore whether the intensity of PSA screening, typically occurring during annual medical check-ups throughout one’s active years, had a notable influence on survival outcomes. Important baseline characteristics by stage and age can be seen in Appendix A. We delved into multivariable models that excluded treatment variables to assess whether potential disparities in survival for localized cancer were shaped by variations in treatment modality (watchful waiting/active surveillance, surgery, radiotherapy) within different occupational groups (Appendix A). Unadjusted (HR) and adjusted hazard ratio (aHR) with 95% confidence interval were calculated. Type-I error was set to 5% where *p*-value less than 0.05 was considered statistically significant.

Data management and statistical analyses are conducted using SAS v9.4 for Windows (SAS Institute Inc., Cary, NC, USA) and SAS Enterprise Guide v5.1. This study was approved by the Institutional Review Boards of the Florida Department of Health and the University of Miami.

## 3. Results

Among a total of 151,681 prostate cancers diagnosed between 2011 and 2014 in Florida, 1058 individuals comprising 84.5% career firefighters and 15.5% volunteers were identified as Florida firefighters through employment and certification records maintained by the State Fire Marshal’s Office (FMO) from 1972 to 2012.

Demographic and clinical characteristics of patients were summarized by occupational groups (FF: firefighters and non-FF: non-firefighters), as detailed in Table 1. For patients diagnosed during 2004–2014, the majority were aged 65–74 years (41.2%), were of white race (81.4%), non-Hispanic (86.6%), insured (92.6%), residing in neighborhoods with poverty levels between 10 and 20% (29.8%), and former or current smokers (35.8%). Most patients were diagnosed with localized SEER stage (81.1%), PSA levels of less than 10 ng/mL at diagnosis (57.0%), and Gleason scores of six or lower (41.3%). Regarding treatment across all stages, 37.7% underwent surgery, 44.1% received radiotherapy, and only 2.7% had records indicating chemotherapy administration. Prostate cancer-related deaths accounted for only 9.3% of the total. Comparing firefighters to non-firefighters diagnosed between 2004 and 2014, firefighters exhibited a higher proportion diagnosed before the age of 65 years (FF: 55.8% vs. non-FF: 37.1%), greater insurance coverage (96.9% vs. 92.6%), a slightly lower percentage of ever-smokers (34.1% vs. 35.8%), and similar proportions living in neighborhoods with poverty levels ≥ 20% (15.2% vs. 15.2%). However, there were fewer Black (9.1% vs. 15.8%) and Hispanic (2.7% vs. 12.5%) patients among firefighters compared to non-firefighters.

Firefighters also exhibited a higher proportion of localized (FF: 84.7% vs. non-FF: 81.1%) and regional (FF: 8.6% vs. non-FF: 7.9%) tumor stages, lower PSA levels at diagnosis (<10 ng/mL) (FF: 65.4% vs. non-FF: 56.9%), and a higher proportion with Gleason scores of six or less (FF: 47.2% vs. non-FF: 41.2%) compared to non-firefighters. Although more firefighters underwent surgery (46.4% vs. 37.6%), fewer received radiotherapy (41.4% vs. 44.1%). Significantly fewer prostate cancer-related deaths were observed among firefighters (5.5% vs. 9.3%) during the study period (*p* < 0.05). Age, mean PSA, and Gleason score distributions by stage of diagnosis are shown in Appendix A.

Cause-specific overall survival at 1-, 3-, 5-, and 10-year intervals for occupational groups is summarized in Table 2. For all patients, survival at these time points were 98.5%, 96.1%, 94.2%, and 90.5%, respectively. Firefighters exhibited significantly higher survival compared to non-firefighters at 3, 5, and 10 years after diagnosis (97.4%, 96.1%, 94.4% vs. 96.0%, 94.2%, 90.5%, respectively).

Kaplan–Meier survival curves based on occupational groups and stage at diagnosis are presented in Figure 1. For all stages combined, a statistically significant better survival for firefighters compared to non-firefighters was observed (*p* < 0.001).

Univariable and multivariable Cox proportional hazard regression models for cause-specific overall survival among prostate cancer patients, with occupational groups as the primary variable, were analyzed by stage at diagnosis. For localized stages, we present all-ages models in Table 3, as there were no major differences by age. However, for regional/distant stages, significant survival differences were noted between those younger than 65 and those aged 65 and older, as demonstrated in Table 4.

In univariable models, firefighters exhibited significantly higher cause-specific survival in localized stages (HR = 0.46; 95% CI: 0.30–0.71; *p* < 0.001), as well as in regional/distant stages for those younger than 65 (HR 0.37; 0.18–0.78), but worse survival for regional/distant stages among those aged 65 and above (HR 1.65; 1.06–2.56).

Three multivariable models (Table 3 and Table 4), featuring occupational groups as the primary variable, were developed to account for differences related to the year of cancer diagnosis, age at cancer diagnosis, race, ethnicity, health insurance, neighborhood socioeconomic status (SES), smoking status, SEER tumor stage when relevant, histology, and treatment received (surgery, radiation therapy, chemotherapy). In localized stages (Table 3), the survival advantage for firefighters observed in univariable analysis was confirmed in multivariable analysis (aHR 0.53; 0.34, 082), a pattern that persisted even after removing treatment variables (surgery, radiotherapy, and chemotherapy) (aHR 0.52; 0.33–0.80) (Appendix A). For regional/distant stages (Table 4), no significant survival differences were observed for those younger than 65, while firefighters aged 65 and above demonstrated a higher risk of death compared to non-firefighters (aHR 1.84; 1.18–2.86). Lastly, in all models, tumor, and serum (PSA) characteristics known to carry a worse prognosis, such as a higher age at diagnosis, undifferentiated tumors, higher PSA levels, and higher Gleason scores, were associated with an elevated risk of death compared to their respective lower counterparts. Surgery and radiotherapy were associated with a lower risk of death in the localized stage, while among regional/distant stages, only surgery had a beneficial effect.

## 4. Discussion

Our study offers valuable insights into the prostate cancer outcomes of firefighters. Notably, our results reveal distinct patterns in this occupational group compared to the general population with the same cancer type. Firefighters diagnosed with prostate cancer tend to be diagnosed at a younger age, often exhibiting more favorable prognostic characteristics. These characteristics include a higher proportion of cases diagnosed at earlier stages, lower PSA levels, and lower Gleason scores. Additionally, firefighters are more likely to undergo surgery or radiotherapy as part of their treatment regimen, which may contribute to the significantly higher five-year survival observed among this group.

One key factor contributing to these findings is the likely heightened level of PSA screening within the firefighting profession [24]. This intensified screening results in a disproportionate number of prostate cancers being detected at localized stages, which positively impacts survival. To account for potential biases introduced by this screening intensity, we conducted stratified analyses based on the stage of diagnosis, distinguishing between localized and regional/distant stages. This approach allowed us to gain a more comprehensive understanding of survival outcomes. Our results indicate that firefighters with localized prostate cancer consistently exhibit better survival outcomes across all age groups when compared to the general population, even after adjustment for all included confounders as well as treatment modalities. However, the result varies for regional/distant stage cancers, particularly in relation to age. Among younger individuals (<65), where indolent cancers are less common, and firefighters likely receive more frequent medical checkups, no significant survival differences were observed between firefighters and non-firefighters. In contrast, among individuals aged 65 or older, firefighters exhibited a less favorable prognosis even after adjusting for confounding factors, including PSA levels, Gleason scores, insurance, treatment, and socioeconomic factors.

Several well-documented epidemiological effects associated with screening, such as lead time bias, length bias, and overdiagnosis, may contribute to our findings [25]. These effects are evident with a higher proportion of localized-stage cancers (more indolent cancers), most likely overdiagnosis where a portion of the detected cancers would not advance during the patient’s lifetime, leading to associated overtreatment, also longer survival times, and possibly more aggressive regional/distant stage cancers among firefighters compared to the general population.

Beyond these findings, it is essential to recognize the unique attributes of firefighters as an occupational group and their interactions with the healthcare system. Firefighters often undergo frequent medical examinations and possess a higher degree of health awareness due to their profession. Many are cross trained as EMT/paramedics, which equips them with medical knowledge that can positively impact screening and treatment engagement, compliance, and navigation within the healthcare system [20].

We could only find one study on this topic involving Norwegian firefighters who were found to experience higher survival compared to the country’s general population, but this difference disappeared after adjusting for age [16]. However, this study did not separate survival outcomes based on the stage of diagnosis. Moreover, differences between the US and Norway, such as higher prostate cancer incidence and mortality rates in Norway [6,26], as well as historically a higher prevalence of PSA testing in the US [27,28], underscore the need for context-specific analysis. Notably, in the US, both firefighters and the general population exhibit more favorable stage distribution, Gleason score distribution, and lower PSA levels.

To gain a more comprehensive understanding of prostate cancer survival disparities between firefighters and non-firefighters, further research is warranted. Future studies should explore potential worse prognoses for regional/distant-stage prostate cancer and assess the impact of localized cancer treatment on the quality of life, particularly among firefighters who more often undergo surgery, which is associated with potential side effects such as impotence and incontinence among others [29].

Our study has several limitations and strengths. One limitation lies in the inclusion of firefighters possibly from out of state who retired in Florida, in the general population group. However, given their relative size, it is unlikely that these individuals would significantly affect the survival experience of the general population. Another limitation arises from changes in PSA screening recommendations over time, which we were unable to account for in our analysis partly due to the relatively low annual numbers of firefighter cases. Additionally, information on occupational carcinogenic exposure or the number of years in a firefighting career was not available in FCDS. Furthermore, cancer registry data inherently entail additional limitations, notably the absence of comprehensive information regarding detailed treatment modalities, specific pharmacological agents administered, and critical clinical details like recurrence status.

Despite these limitations, our study is the first of its kind to provide population-based epidemiological insights into cause-specific overall survival from prostate cancer among firefighters, both career and volunteer, in Florida. It stands out by capturing the real-life experiences of firefighters and a diverse general population, avoiding the selection biases often associated with hospital- or National Cancer Database (NCDB)-based studies [30].

## 5. Conclusions

Our study reveals that firefighters in Florida experience significantly higher prostate cancer survival compared to non-firefighters, which may be attributable to a likely higher frequency of PSA testing in this occupational group. Additionally, our study underscores the uniqueness of firefighters as an occupational group with heightened medical scrutiny and awareness. This heightened awareness, coupled with favorable stage distribution contributes to their superior survival outcomes. However, disparities emerge for regional/distant stage cancers, particularly among older firefighters, suggesting a possible interplay between earlier detection, stage shifting due to PSA and the presence of unaccounted factors affecting survival outcomes in this subgroup.

Comprehensive real-life analyses, like this population-based study focused on a specific occupational group, are crucial for gaining a better understanding of the impact of PSA on survival outcomes in the population. Moreover, these findings emphasize the need for further investigation into disparities in more advanced stages of prostate cancer between firefighters and non-firefighters. Future epidemiological studies are crucial for understanding and linking occupational and environmental exposure data to cancer cohort studies in this unique population.

## Figures and Tables

**Figure 1 cancers-16-01305-f001:**
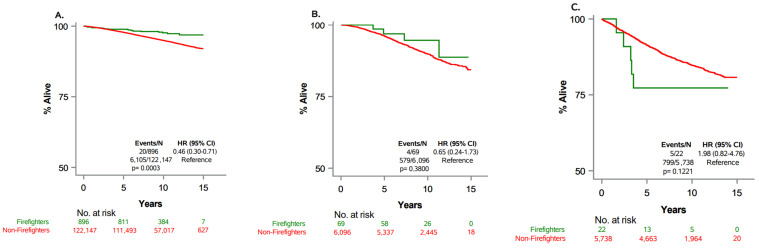
Overall survival by occupation in sub samples. (**A**). In localized stage, (**B**). in regional/distant stage and age < 65, (**C**). in regional/distant stage and age ≥ 65.

**Table 1 cancers-16-01305-t001:** Demographics and clinical characteristics of prostate cancer patients by firefighting occupation status: Florida Cancer Data System (2004–2014).

	Total		Occupation				
			Firefighters		Non-Firefighters		
	N	Col%	N	Col%	N	Col%	*p*-Value ^1^
All	151,681	100	1058	100	150,623	100	
**Year of diagnosis**							
2004–2009	87,679	57.8	577	54.5	87,102	57.8	0.031
2010–2014	64,002	42.2	481	45.5	63,521	42.2	
**Age at diagnosis**							
18–54	13,170	8.7	186	17.6	12,984	8.6	<0.001
55–64	43,310	28.6	404	38.2	42,906	28.5	
65–74	62,445	41.2	337	31.9	62,108	41.2	
75+	32,756	21.6	131	12.4	32,625	21.7	
**Mean (SD)**	67.3 (9.2)		63.2 (9.3)		67.3 (9.2)		
**Median (Min, Max)**	67 (29, 104)		63 (36, 96)		67 (29, 104)		
**Race**							
White	123,519	81.4	936	88.5	122,583	81.4	<0.001
Black	23,910	15.8	96	9.1	23,814	15.8	
Other/Unknown	4252	2.8	26	2.5	4226	2.8	
**Ethnicity**							
Non-Hispanic	131,325	86.6	1015	95.9	130,310	86.5	<0.001
Hispanic	18,878	12.4	29	2.7	18,849	12.5	
Unknown	1478	1	14	1.3	1464	1	
**Insurance**							
Uninsured/Unknown	11,230	7.4	33	3.1	11,197	7.4	<0.001
Insured	140,451	92.6	1025	96.9	139,426	92.6	
**SES–% poverty level**							
20–100% poverty	22,981	15.2	161	15.2	22,820	15.2	0.06
10–<20% poverty	45,205	29.8	387	36.6	44,818	29.8	
5–<10% poverty	41,351	27.3	296	28	41,055	27.3	
0–<5% poverty	20,876	13.8	162	15.3	20,714	13.8	
Unknown	21,268	14	52	4.9	21,216	14.1	
**Cigarette use**							
Never	55,849	36.8	434	41	55,415	36.8	0.034
History	13,769	9.1	96	9.1	13,673	9.1	
Current	40,494	26.7	264	25	40,230	26.7	
Unknown	41,569	27.4	264	25	41,305	27.4	
**SEER stage**							
Localized	123,043	81.1	896	84.7	122,147	81.1	0.004
Regional	11,925	7.9	91	8.6	11,834	7.9	
Distant	6393	4.2	24	2.3	6369	4.2	
Unknown	10,320	6.8	47	4.4	10,273	6.8	
**Grade**							
Well-differentiated	7116	4.7	57	5.4	7059	4.7	0.051
Moderately differentiated	66,074	43.6	517	48.9	65,557	43.5	
Poorly differentiated	58,639	38.7	391	37	58,248	38.7	
Undifferentiated	919	0.6	10	0.9	909	0.6	
Unknown/not stated	18,933	12.5	83	7.8	18,850	12.5	
**PSA**							
<10 ng/mL	86,402	57	692	65.4	85,710	56.9	<0.001
10–20 ng/mL	14,821	9.8	94	8.9	14,727	9.8	
>20 ng/mL	13,473	8.9	66	6.2	13,407	8.9	
Unknown	36,985	24.4	206	19.5	36,779	24.4	
**Mean (SD)**	12.3 (19.3)		9.5 (14.8)		12.4 (19.3)		
**Median (Min, Max)**	6.0 (0.1, 98)		5.4 (0.1, 98.0)		6.0 (0.1, 98.0)		
**Gleason scores**							
≤6	62,569	41.3	499	47.2	62,070	41.2	0.023
7	53,909	35.5	363	34.3	53,546	35.5	
≥8	10,723	7.1	91	8.6	10,632	7.1	
Unknown	24,480	16.1	105	9.9	24,375	16.2	
**Receipt of Surgery**							
No	93,170	61.4	560	52.9	92,610	61.5	<0.001
Yes	57,197	37.7	491	46.4	56,706	37.6	
Unknown	1314	0.9	<10	*	1307	0.9	
**Receipt of Radiotherapy**							
No	78,423	51.7	598	56.5	77,825	51.7	0.016
Yes	66,826	44.1	438	41.4	66,388	44.1	
Unknown	6432	4.2	22	2.1	6410	4.3	
**Receipt of Chemotherapy**							
No	144,923	95.5	1040	98.3	143,883	95.5	<0.001
Yes	4160	2.7	10	0.9	4150	2.8	
Unknown	2598	1.7	<10	*	2590	1.7	
**Vital Status**							
Alive/Dead—other cause	137,621	90.7	1000	94.5	136,621	90.7	NA
Dead—primary diagnosed	14,060	9.3	58	5.5	14,002	9.3	

*: Proportions not shown as N is less than 10; SES: socio-economic status; SES: socioeconomic status reported as the percent poverty level of the patients’ neighborhood at the time of cancer diagnosis. PSA: prostatic-specific antigen; SEER: Surveillance, Epidemiology, and End Results (SEER) Program; SD: standard deviation; Min: minimum; Max: maximum. Vital status and follow-up duration were verified using records from the Florida Office of Vital Statistics and the National Death Index (2004–2019). ^1^ Chi-square test. NA: not applicable.

**Table 2 cancers-16-01305-t002:** One-, three-, five-, and ten-year survival proportions for firefighters and non-firefighters.

Variable	N	Survival Time Proportions with 95% Confidence Interval			
		1 Year (%)	3 Years (%)	5 Years (%)	10 Years (%)
**All**	151,681	98.5 (98.4–98.5)	96.1 (96.0–96.2)	94.2 (94.1–94.3)	90.5 (90.3–90.6)
**Occupation**					
Firefighters	1058	98.9 (98.0–99.4)	97.4 (96.3–98.2)	96.1 (94.7–97.1)	94.4 (92.7–95.7)
Non-Firefighters	150,623	98.5 (98.4–98.5)	96.0 (95.9–96.1)	94.2 (94.1–94.3)	90.5 (90.3–90.6)

Survival from prostate cancer was studied using the elapsed time in days from the date of cancer diagnosis to the earliest of the two dates: date of death or 31 December 2019, for alive patients.

**Table 3 cancers-16-01305-t003:** Determinants of localized stage prostate cancer survival, Florida 2004–2014.

Variable	Category	N	%	Univariable Model (UVA)		Multivariable Model (MVA)	
				HR (95%CI)	*p*-Value	aHR (95%CI)	*p*-Value
**Occupation**	Non-Firefighters	896	0.7	Ref		Ref	
	Firefighters	122,147	99.3	0.46 (0.30, 0.71)	<0.001	0.53 (0.34, 0.82)	0.004
**Year of diagnosis**	2004–2009	72,067	58.6	Ref		Ref	
	2010–2014	50,976	41.4	0.91 (0.86, 0.97)	0.004	0.79 (0.74, 0.85)	<0.001
**Age at diagnosis**	18–54	10,753	8.7	Ref		Ref	
**(years)**	55–64	35,340	28.7	1.53 (1.32, 1.77)	<0.001	1.39 (1.21, 1.61)	<0.001
	65–74	52,259	42.5	2.31 (2.01, 2.65)	<0.001	1.99 (1.73, 2.29)	<0.001
	75+	24,691	20.1	4.63 (4.03, 5.32)	<0.001	3.20 (2.78, 3.69)	<0.001
**Race**	White	100,592	81.8	Ref		Ref	
	Black	19,054	15.5	1.04 (0.97, 1.11)	0.272	1.00 (0.92, 1.07)	0.93
	Other/Unknown	3397	2.8	0.68 (0.56, 0.82)	<0.001	0.74 (0.61, 0.89)	0.002
**Ethnicity**	Non-Hispanic	106,632	86.7	Ref		Ref	
	Hispanic	15,184	12.3	0.90 (0.84, 0.98)	0.014	0.96 (0.87, 1.05)	0.325
	Unknown	1227	1.0	0.54 (0.39, 0.77)	<0.001	0.70 (0.49, 0.99)	0.045
**Insurance**	Uninsured/Unknown	6941	5.6	Ref		Ref	
	Insured	116,102	94.4	0.85 (0.76, 0.94)	0.001	0.74 (0.66, 0.83)	<0.001
**SES–** **% poverty level**	20–100% poverty	18,334	14.9	Ref		Ref	
	10–<20% poverty	36,580	29.7	0.86 (0.80, 0.93)	<0.001	0.92 (0.85, 0.99)	0.03
	5–<10% poverty	34,056	27.7	0.78 (0.72, 0.84)	<0.001	0.84 (0.77, 0.91)	<0.001
	0–<5% poverty	17,495	14.2	0.70 (0.64, 0.77)	<0.001	0.80 (0.72, 0.88)	<0.001
	Unknown	16,578	13.5	0.71 (0.65, 0.78)	<0.001	0.76 (0.69, 0.85)	<0.001
**Cigarette use**	Never	46,041	37.4	Ref		Ref	
	History	10,837	8.8	1.53 (1.40, 1.67)	<0.001	1.60 (1.46, 1.75)	<0.001
	Current	33,579	27.3	1.25 (1.18, 1.34)	<0.001	1.14 (1.06, 1.21)	<0.001
	Unknown	32,586	26.5	1.36 (1.27, 1.45)	<0.001	1.17 (1.10, 1.26)	<0.001
**Grade**	Well-differentiated	6665	5.4	Ref		Ref	
	Moderately differentiated	61,503	50.0	1.06 (0.87, 1.28)	0.573	0.99 (0.82, 1.20)	0.93
	Poorly differentiated	46,124	37.5	2.87 (2.37, 3.47)	<0.001	1.39 (1.13, 1.72)	0.002
	Undifferentiated	485	0.4	6.03 (4.50, 8.08)	<0.001	2.33 (1.72, 3.17)	<0.001
	Unknown/not stated	8266	6.7	3.12 (2.55, 3.82)	<0.001	1.25 (1.00, 1.56)	0.054
**PSA**	<10 ng/mL	78,418	63.7	Ref		Ref	
	10–20 ng/mL	12,288	10.0	2.12 (1.96, 2.28)	<0.001	1.55 (1.44, 1.67)	<0.001
	>20 ng/mL	7941	6.5	3.37 (3.13, 3.63)	<0.001	2.25 (2.08, 2.43)	<0.001
	Unknown	24,396	19.8	1.66 (1.56, 1.77)	<0.001	1.56 (1.46, 1.68)	<0.001
**Gleason score**	≤6	58,699	47.7	Ref		Ref	
	7	44,865	36.5	2.63 (2.47, 2.80)	<0.001	1.78 (1.61, 1.96)	<0.001
	≥8	7077	5.8	7.02 (6.42, 7.67)	<0.001	4.17 (3.66, 4.76)	<0.001
	Unknown	12,402	10.1	3.35 (3.10, 3.63)	<0.001	2.20 (1.95, 2.47)	<0.001
**Receipt of Surgery**	No	77,677	63.1	Ref		Ref	
	Yes	45,026	36.6	0.57 (0.54, 0.60)	<0.001	0.59 (0.55, 0.64)	<0.001
	Unknown	340	0.3	1.10 (0.74, 1.63)	0.629	0.88 (0.59, 1.31)	0.531
**Receipt of Radiotherapy**	No	57,796	47.0	Ref		Ref	
	Yes	60,557	49.2	1.09 (1.04, 1.15)	<0.001	0.72 (0.68, 0.77)	<0.001
	Unknown	4690	3.8	0.71 (0.60, 0.84)	<0.001	0.39 (0.31, 0.49)	<0.001
**Receipt of Chemotherapy**	No	118,841	96.6	Ref		Ref	
	Yes	2778	2.3	0.93 (0.78, 1.11)	0.414	1.22 (0.94, 1.56)	0.131
	Unknown	1424	1.2	1.31 (1.08, 1.58)	0.005	1.63 (1.35, 1.98)	<0.001

Ref: Reference group; SES: socio-economic status; SES: socioeconomic status reported as the percent poverty level of the patients’ neighborhood at the time of cancer diagnosis. PSA: prostatic-specific antigen; SD: standard deviation; Min: minimum; Max: maximum; HR: hazard ratio; aHR: adjusted hazard ratio; 95%CI: 95% confidence interval.

**Table 4 cancers-16-01305-t004:** Determinants of regional/distant stage prostate cancer survival by age group <65 and ≥65 years of age, Florida 2004–2014.

Variable	Category	Age Group (Years)											
		<65 Years, *n* = 8101						≥65 Years, *n* = 10,217					
		N	%	Univariate Model (UVA)		Multivariable Model (MVA)		N	%	Univariate Model (UVA)		Multivariable Model (MVA)	
				HR (95%CI)	*p*-Value	aHR (95%CI)	*p*-Value			HR (95%CI)	*p*-Value	aHR (95%CI)	*p*-Value
**Occupation**	Non-Firefighters	8025	99.1	Ref		Ref		10,178	99.6	Ref		Ref	
	Firefighters	76	0.9	0.37 (0.18, 0.78)	0.009	0.64 (0.30, 1.34)	0.233	39	0.4	1.65 (1.06, 2.56)	0.026	1.84 (1.18–2.86)	0.007
**Year of diagnosis**	2004–2009	4292	53	Ref		Ref		5010	49	Ref		Ref	
	2010–2014	3809	47	1.15 (1.05, 1.27)	0.004	0.92 (0.82, 1.04)	0.175	5207	51	1.07 (1.01, 1.15)	0.033	1.03 (0.95, 1.11)	0.484
**Age at diagnosis (years)**	18–54	2010	24.8	Ref		Ref		-	-	NA		NA	
	55–64	6091	75.2	1.04 (0.94, 1.16)	0.418	0.95 (0.85, 1.06)	0.355	-	-	NA		NA	
	65–74	-	-	NA		NA		6524	63.9	Ref		Ref	
	75+	-	-	NA		NA		3693	36.1	2.69 (2.52, 2.87)	<0.001	1.30 (1.21, 1.39)	<0.001
**Race**	White	6122	75.6	Ref		Ref		8564	83.8	Ref		Ref	
	Black	1751	21.6	1.42 (1.28, 1.57)	<0.001	0.93 (0.83, 1.05)	0.234	1392	13.6	1.32 (1.21, 1.44)	<0.001	0.96 (0.87, 1.05)	0.379
	Other/Unknown	228	2.8	0.52 (0.35, 0.76)	0.001	0.59 (0.40, 0.88)	0.01	261	2.6	0.63 (0.49, 0.81)	<0.001	0.74 (0.57, 0.95)	0.018
**Ethnicity**	Non-Hispanic	7022	86.7	Ref		Ref		8825	86.4	Ref		Ref	
	Hispanic	1003	12.4	1.02 (0.89, 1.17)	0.747	0.87 (0.74, 1.01)	0.059	1340	13.1	0.84 (0.76, 0.93)	<0.001	0.95 (0.85, 1.07)	0.394
	Unknown	76	0.9	0.52 (0.35, 0.76)	<0.001	0.68 (0.36, 1.28)	0.235	52	0.5	0.75 (0.45, 1.24)	0.257	0.91 (0.55, 1.53)	0.729
**Insurance**	Uninsured/Unknown	958	11.8	Ref		Ref		669	6.5	Ref		Ref	
	Insured	7143	88.2	0.54 (0.48, 0.60)	<0.001	1.06 (0.94, 1.20)	0.332	9548	93.5	0.89 (0.79, 1.01)	0.068	1.09 (0.96, 1.24)	0.193
**SES—% poverty level**	20–100% poverty	1448	17.9	Ref		Ref		1675	16.4	Ref		Ref	
	10–<20% poverty	2462	30.4	0.81 (0.71, 0.91)	<0.001	0.98 (0.86, 1.11)	0.75	3175	31.1	0.85 (0.77, 0.93)	<0.001	0.99 (0.90, 1.09)	0.875
	5–<10% poverty	1970	24.3	0.60 (0.52, 0.69)	<0.001	0.88 (0.76, 1.01)	0.076	2731	26.7	0.79 (0.72, 0.87)	<0.001	0.99 (0.90, 1.10)	0.911
	0–<5% poverty	1082	13.4	0.52 (0.44, 0.62)	<0.001	0.84 (0.70, 1.00)	0.048	1225	12	0.70 (0.62, 0.79)	<0.001	0.91 (0.81, 1.04)	0.166
	Unknown	1139	14.1	0.74 (0.63, 0.86)	<0.001	0.86 (0.73, 1.01)	0.073	1411	13.8	0.66 (0.59, 0.74)	<0.001	0.85 (0.75, 0.97)	0.016
**Cigarette use**	Never	2946	36.4	Ref		Ref		3702	36.2	Ref		Ref	
	History	1239	15.3	1.72 (1.52, 1.96)	<0.001	1.21 (1.06, 1.37)	0.005	884	8.7	1.11 (0.99, 1.25)	0.087	1.18 (1.04, 1.33)	0.009
	Current	1619	20	1.13 (1.00, 1.29)	0.058	0.98 (0.86, 1.12)	0.736	2907	28.5	1.05 (0.97, 1.14)	0.213	1.08 (0.99, 1.17)	0.079
	Unknown	2297	28.4	1.12 (0.99, 1.26)	0.062	1.00 (0.89, 1.13)	0.981	2724	26.7	1.11 (1.03, 1.21)	0.009	1.06 (0.97, 1.15)	0.193
**SEER Stage**	Regional	6165	76.1	Ref		Ref		5760	56.4	Ref		Ref	
	Distant	1936	23.9	13.19 (11.95, 14.57)	<0.001	4.93 (4.25, 5.71)	<0.001	4457	43.6	7.84 (7.25, 8.48)	<0.001	4.46 (4.02, 4.94)	<0.001
**Grade**	Well-differentiated	75	0.9	Ref		Ref		78	0.8	Ref		Ref	
	Moderately differentiated	1810	22.3	0.62 (0.25, 1.53)	0.3	1.11 (0.45, 2.73)	0.826	1552	15.2	0.34 (0.22, 0.52)	<0.001	0.64 (0.41, 1.00)	0.053
	Poorly differentiated	5266	65	3.06 (1.27, 7.36)	0.013	2.65 (1.09, 6.45)	0.031	6007	58.8	1.10 (0.72, 1.68)	0.654	1.17 (0.76, 1.81)	0.467
	Undifferentiated	167	2.1	6.93 (2.81, 17.11)	<0.001	4.04 (1.62, 10.09)	0.003	196	1.9	2.00 (1.26, 3.16)	0.003	1.67 (1.04, 2.67)	0.033
	Unknown/not stated	783	9.7	10.51 (4.35, 25.38)	<0.001	2.84 (1.17, 6.93)	0.022	2384	23.3	2.83 (1.86, 4.32)	<0.001	0.92 (0.60, 1.42)	0.711
**PSA**	<10 ng/mL	3655	45.1	Ref		Ref				Ref		Ref	
	10–20 ng/mL	1028	12.7	2.05 (1.73, 2.43)	<0.001	1.29 (1.08, 1.54)	0.004	3364	32.9	1.98 (1.74, 2.26)	<0.001	1.22 (1.07, 1.40)	0.003
	>20 ng/mL	1825	22.5	7.36 (6.53, 8.29)	<0.001	1.58 (1.38, 1.82)	<0.001	1218	11.9	4.77 (4.34, 5.24)	<0.001	1.50 (1.35, 1.67)	<0.001
	Unknown	1593	19.7	2.32 (2.00, 2.68)	<0.001	1.10 (0.94, 1.29)	0.223	3215	31.5	2.92 (2.63, 3.23)	<0.001	1.29 (1.15, 1.44)	<0.001
**Gleason score**	≤6	1578	19.5	Ref		Ref		1192	11.7	Ref		Ref	
	7	4044	49.9	3.63 (2.88, 4.58)	<0.001	1.52 (1.17, 1.96)	0.002	4108	40.2	2.88 (2.37, 3.48)	<0.001	1.43 (1.15, 1.78)	0.001
	≥8	1306	16.1	10.51 (8.27, 13.36)	<0.001	1.82 (1.37, 2.41)	<0.001	2069	20.3	5.51 (4.53, 6.69)	<0.001	1.61 (1.28, 2.03)	<0.001
	Unknown	1173	14.5	15.11(11.96,19.10)	<0.001	1.75 (1.32, 2.33)	<0.001	2848	27.9	10.27 (8.51,12.39)	<0.001	2.12 (1.68, 2.68)	<0.001
**Receipt of Surgery**	No	2364	29.2	Ref		Ref		5441	53.3	Ref		Ref	
	Yes	5723	70.6	0.11 (0.10, 0.12)	<0.001	0.47 (0.41, 0.55)	<0.001	4749	46.5	0.23 (0.22, 0.25)	<0.001	0.82 (0.75, 0.91)	<0.001
	Unknown	14	0.2	1.86 (1.03, 3.37)	0.041	1.75 (0.93, 3.30)	0.081	27	0.3	0.92 (0.54, 1.59)	0.773	0.98 (0.56, 1.71)	0.944
**Receipt of Radiotherapy**	No	6047	74.6	Ref		Ref		7327	71.7	Ref		Ref	
	Yes	1702	21	1.78 (1.61, 1.96)	<0.001	1.13 (1.02, 1.25)	0.019	2491	24.4	0.91 (0.85, 0.99)	0.019	1.00 (0.92, 1.08)	0.935
	Unknown	352	4.3	0.81 (0.62, 1.06)	0.127	0.94 (0.71, 1.24)	0.639	399	3.9	0.47 (0.37, 0.58)	<0.001	0.64 (0.50, 0.81)	<0.001
**Receipt of Chemotherapy**	No	7534	93	Ref		Ref		9481	92.8	Ref		Ref	
	Yes	456	5.6	3.29 (2.87, 3.77)	<0.001	1.31 (1.14, 1.52)	<0.001	553	5.4	1.86 (1.65, 2.09)	<0.001	1.30 (1.15, 1.46)	<0.001
	Unknown	111	1.4	1.74 (1.25, 2.41)	<0.001	1.32 (0.94, 1.88)	0.114	183	1.8	0.99 (0.78, 1.27)	0.961	1.00 (0.78, 1.29)	0.997

Ref: Reference group; SES: socio-economic status; SES: socioeconomic status reported as the percent poverty level of the patients’ neighborhood at the time of cancer diagnosis. PSA: prostatic-specific antigen; SEER: Surveillance, Epidemiology, and End Results (SEER) Program; SD: standard deviation; Min: minimum; Max: maximum; HR: hazard ratio; aHR: adjusted hazard ratio; 95%CI: 95% confidence interval.

## Data Availability

The datasets presented in this article are not readily available due to strict confidentiality agreements between the University of Miami and the Florida Department of Health, Florida Cancer Data System, and Florida Fire Marshalls Office. This study was approved by the Institutional Review Boards of the Florida Department of Health and the University of Miami. A waiver of informed consent was granted given that cancer data is a reportable event for the purposes of cancer surveillance. Requests to access the datasets should be directed to the Florida Department of Health, health@flhealth.gov.

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
