# Peer review of "Distinct Prostate Cancer Survival Outcomes in Firefighters: A Population-Based Study"

_cancers, 2024, doi:10.3390/cancers16071305_

Round 1
Reviewer 1 Report
Comments and Suggestions for Authors
The objective of this study is to examine and compare the characteristics and survival outcomes of firefighters and non-firefighters.
Despite the fact that the study focused on patients diagnosed with prostate cancer prior to 2014, a significant proportion of these patients had low-risk disease and were mostly treated with active treatment. The matter of overdiagnosis and overtreatment should be addressed in discussion.
In the regional/distant stage, the patients were categorized into two age groups: those younger than 65 years and those older than 65 years. However, the group of patients aged 65 years and above consisted of just 22 individuals. The size of this sample is insufficient for conducting a thorough statistical analysis and drawing meaningful conclusions.
From my perspective, the rationale behind utilizing chemotherapy as the exclusive therapeutic approach for individuals with localized disease remains unclear. Due to the limited number of patients getting this type of treatment, it is advisable to exclude this data from the statistical analysis.
The regional/distant stage exhibits significant heterogeneity, ranging from node-positive, to recurrent, to de novo metastatic disease. Therapy for these stages typically involves a multimodal approach with the use of various therapy lines. Therefore, the factors used for localized disease are insufficient. Furthermore, there are other confounding factors that cannot be measured using the SEER dataset. Additionally, as previously mentioned, the analyses are limited due to the small sample size.
The unmentioned major limitations is to the inherent constraints associated with utilizing the SEER dataset, which necessitates further elaboration.
Author Response
- Despite the fact that the study focused on patients diagnosed with prostate cancer prior to 2014, a significant proportion of these patients had low-risk disease and were mostly treated with active treatment. The matter of overdiagnosis and overtreatment should be addressed in the discussion.
Thank you for your valuable feedback on our manuscript. We acknowledge the concerns regarding overdiagnosis and overtreatment, particularly given the focus on patients who exhibited low-risk disease and were predominantly subjected to active treatment. The issue of overdiagnosis stemming from the widespread use of PSA testing in the US, compounded by the heightened health awareness among specific occupational groups such as firefighters, leading to increased health check-ups and referrals, is addressed in our discussion. Additionally, we delve into the matter of overtreatment, highlighting the notably elevated rates of surgery for localized cancer evident throughout our text, with specific emphasis on paragraphs 3 and 6 of the discussion section. We sincerely appreciate your attention to these critical points.
- In the regional/distant stage, the patients were categorized into two age groups: those younger than 65 years and those older than 65 years. However, the group of patients aged 65 years and above consisted of just 22 individuals. The size of this sample is insufficient for conducting a thorough statistical analysis and drawing meaningful conclusions.
Thank you for your insightful observation. While we acknowledge that the subgroup of individuals aged 65 years and above among firefighters diagnosed at the regional/distant stage comprises just 22 individuals, it's crucial to note that despite this relatively small sample size, our analysis yielded statistically significant results. This suggests that while the sample size may be limited, the findings remain robust. It's important to clarify that this study is population-based (not sample based), encompassing the entirety of firefighters and the population with prostate cancer in a state with a population of 22 million. Therefore, we are not dealing with a traditional sample, but rather the complete set of cases within this demographic. This distinction underscores the comprehensiveness of our data, despite the smaller subgroup size. Furthermore, we call for further research from various US states and different regions worldwide to validate and expand upon our findings. Additionally, we contextualize our results by considering factors such as the likelihood of higher PSA testing among firefighters, which may influence the stage distribution observed in our study.
- From my perspective, the rationale behind utilizing chemotherapy as the exclusive therapeutic approach for individuals with localized disease remains unclear. Due to the limited number of patients getting this type of treatment, it is advisable to exclude this data from the statistical analysis.
Indeed, we concur that it's improbable for any patient to receive chemotherapy as the sole treatment for localized cancer. The variables introduced in the multivariable method merely indicate whether these treatments were administered or not, reflecting the constraints of cancer registry data. It's important to note that surgery, radiotherapy, and chemotherapy are not mutually exclusive variables. The combination of yes to chemotherapy, no to surgery and no to radiotherapy for patients diagnosed in localized stage in our study showed exactly 1 firefighter and 100 prostate cancers in the general population. Given the larger numbers we ran the model without these 101 subjects an unsurprisingly the model was not altered.
- The regional/distant stage exhibits significant heterogeneity, ranging from node-positive, to recurrent, to de novo metastatic disease. Therapy for these stages typically involves a multimodal approach with the use of various therapy lines. Therefore, the factors used for localized disease are insufficient.
This study is designed as a population-based epidemiological investigation rather than a clinical one. Therefore, it prioritizes inclusivity by encompassing subjects from various healthcare settings such as hospitals, clinics, and private offices, aiming to minimize selection biases. However, this inclusivity comes at the expense of limited clinical information and treatment detail. Unfortunately, the specific treatment data suggested by the reviewer, including the nuances of therapy lines for different stages, are unavailable in cancer registry records. Moreover, there is no info on recurrences, all cases in cancer registries are de novo diagnoses.
- Furthermore, there are other confounding factors that cannot be measured using the SEER dataset. Additionally, as previously mentioned, the analyses are limited due to the small sample size. The unmentioned major limitations are to the inherent constraints associated with utilizing the SEER dataset, which necessitates further elaboration.
We agree with the reviewer and added these to the limitations.
Reviewer 2 Report
Comments and Suggestions for Authors
With this work author compared the survival of 1,058 firefighter diagnosed for prostate cancer to patients with prostate cancer in the general population, finding significantly better five-year survival in firefighters. Authors tried to identify the motivations of this differences and they found out that the reason is that firefighters undergo more severe and frequent medical tests so they are diagnosed earlier, have more localized cancers, and undergo surgery more often. More studies are warranted to assess the impact of a higher rate of surgical treatment on firefighters' quality of life.
My suggestion:
English language should be revised and corrected, the text is not clear in many points.
Many references from the bibliography are older than ten years, please cite up-to-date works to give findings stronger evidences
This is a retrospective observational study, prone to bias in data collections. A randomized clinical trial would give findings a stronger scientific weight. More studies are required
Survival can be influenced by many factors, among which comorbidities or therapeutic strategies chosen to face the malignancy. No mention in this work is done about external factors that can influence survival so at this regard I can kindly suggest the analysis of this two works: https://pubmed.ncbi.nlm.nih.gov/32570240/
https://pubmed.ncbi.nlm.nih.gov/37419854/
Comments on the Quality of English LanguageMinor editing
Author Response
- English language should be revised and corrected, the text is not clear in many points.
Thank you for your constructive feedback. We appreciate your input. If you could provide more detailed suggestions or examples, we would be better equipped to enhance the clarity and coherence of the text.
- Many references from the bibliography are older than ten years, please cite up-to-date works to give findings stronger evidences.
Respectfully, it's worth noting that out of the 34 references cited, only four are older than a decade. These references primarily pertain to methodologies or offer insights into PSA prevalence in Norway, where limited alternative studies on the topic are available. Importantly, these older references are not substantially influenced by recent discoveries within the field, reinforcing their continued relevance to our study.
- This is a retrospective observational study, prone to bias in data collections. A randomized clinical trial would give findings a stronger scientific weight. More studies are required.
We concur that randomized trials represent the gold standard in scientific research, particularly when feasible. However, in this particular case, the nature of the research question inherently lends itself to observational methodologies. Given the constraints and ethical considerations involved, we are uncertain how a randomized trial would be suitable for adequately addressing the specific research question at hand.
- Survival can be influenced by many factors, among which comorbidities or therapeutic strategies chosen to face the malignancy. No mention in this work is done about external factors that can influence survival so at this regard I can kindly suggest the analysis of this two works: https://pubmed.ncbi.nlm.nih.gov/32570240/ and https://pubmed.ncbi.nlm.nih.gov/37419854/.
We appreciate the reviewer's contribution of these insightful articles. The references provided address the treatment of metastatic prostate cancer, a topic of utmost significance. However, it is essential to note that the applicability of these references to our population-based study is limited. Our study does not encompass data on specific pharmacological agents utilized for treatment, rendering the relevance of these articles somewhat diminished within the context of our research focus.
Reviewer 3 Report
Comments and Suggestions for Authors
1. In Figure 1, you must indicate the confidence interval.
2. Table 1 does not show p-values and it is unclear whether there are significant differences between the samples. In particular, in my opinion, subgroups differ by race and ethnicity.
3. Survival rates are calculated based on the date of death or up to December 31, 2019 for living patients. Were there any patients lost to follow-up? Why is the last monitoring date not used in the analysis?
4. There is no data on concomitant diseases, which is especially important when analyzing age. Apparently, older firefighters have more pathologies associated with cancer, which may contribute to an increase in mortality.
5. In tables 3 and 4, it is necessary to add for each subgroup the number of people in it, since statistics were previously given only for the group as a whole.
6. How often is PCA determined in a group of firefighters? What about the population as a whole? Or do firefighters have additional medical examinations, including the prostate?
Author Response
We would like to thank you Reviewer #3 for his/her/theirs valuable feedback on our manuscript.
- In Figure 1, you must indicate the confidence interval.
Certainly, Figures 1A, 1B, and 1C all feature the confidence interval for the hazard ratios comparing firefighters to non-firefighters, located in the lower right corner.
- Table 1 does not show p-values and it is unclear whether there are significant differences between the samples. In particular, in my opinion, subgroups differ by race and ethnicity.
Thank you very much for this suggestion. P-values have now been added to table 1. Indeed, the racial and ethnic proportions are different between firefighters and non-firefighters. All these variables are adjusted for in the final multivariable models which would take care of these differences.
- Survival rates are calculated based on the date of death or up to December 31, 2019 for living patients. Were there any patients lost to follow-up? Why is the last monitoring date not used in the analysis?
This is a population-based study using data from the Florida Cancer Data System the statewide cancer registry. FCDS is affiliated with the National Program of Cancer Registries (NPCR,) which unlike the SEER Program, conducts passive follow-up (mortality linkage only). This is different from active follow-up for all cancer cases as in SEER in which a date of last alive contact is sought for at least 95% of all cases. See (Pinheiro et al 2014). As per North American Central Cancer Registries Association procedures, in the case of passive follow-up, the date of last (most recent) mortality linkage should be used for all alive patients in survival calculations, In this case, all cancer cases were checked through passive follow-up with US mortality files inclusive of all deaths and respective cause of death for all years up to the end of 2019.
- There is no data on concomitant diseases, which is especially important when analyzing age. Apparently, older firefighters have more pathologies associated with cancer, which may contribute to an increase in mortality.
We totally agree with the reviewer. Precisely because of that, and to reduce the effect of unmeasured comorbidities that may, by themselves, increase mortality, we chose to study cause-specific survival rather than observed survival. This is especially important because of the skewed distribution of prostate cancer towards old age when men can die of many other concurrent causes.
- In tables 3 and 4, it is necessary to add for each subgroup the number of people in it, since statistics were previously given only for the group as a whole.
Your point is well-taken. Subgroup sizes have now been added to Tables 3 and 4, providing a more comprehensive understanding of the statistics within each subgroup.
- How often is PCA determined in a group of firefighters? What about the population as a whole? Or do firefighters have additional medical examinations, including the prostate?
Firefighters undergo occupational physical exams following the National Fire Protection Association (NFPA) 1582 standard that contains a concise list of requirements for medical testing and physical examinations. These exams should be done when a firefighter joins a department, and each year thereafter. In contrast, the general population is not always subject to such routine examinations. The list of exams for firefighters includes items such as: a physical examination, chest x-ray, and cancer screening including PSA testing. Regrettably, data on the prevalence of PSA testing among firefighters are not readily available. While recommendations for PSA testing in the US have evolved over time and are typically made collaboratively between patients and physicians, its widespread use is driven by concerns over potential medical-legal ramifications if prostate cancer is detected too late. Consequently, in the US there is a relatively high prevalence of biannual PSA testing (15% in those aged 40-55, 42% 55-69, 50% in those aged 70 or more) (Merrill et al. 2022) and in global terms as a result, there is a high incidence of prostate cancer (Global Cancer Observatory, Accessed December 2023), even when comparing rates by race separately by race.
Round 2
Reviewer 1 Report
Comments and Suggestions for Authors
No further comment
Author Response
Thank you for reviewing our manuscript.
Reviewer 2 Report
Comments and Suggestions for Authors
Authors have to update references according to reviewer's suggestions.
Comments on the Quality of English LanguageModerate editing.
Author Response
- English language should be revised and corrected; the text is not clear in many points.
Thank you for your constructive feedback. We appreciate your input. If you could provide more detailed suggestions or examples, we would be better equipped to enhance the clarity and coherence of the text.
- Many references from the bibliography are older than ten years, please cite up-to-date works to give findings stronger evidence.
Respectfully, it's worth noting that out of the 34 references cited, only four are older than a decade. These references primarily pertain to methodologies or offer insights into PSA prevalence in Norway, where limited alternative studies on the topic are available. Importantly, these older references are not substantially influenced by recent discoveries within the field, reinforcing their continued relevance to our study.
- This is a retrospective observational study, prone to bias in data collections. A randomized clinical trial would give findings a stronger scientific weight. More studies are required.
We concur that randomized trials represent the gold standard in scientific research, particularly when feasible. However, in this particular case, the nature of the research question inherently lends itself to observational methodologies. Given the constraints and ethical considerations involved, we are uncertain how a randomized trial would be suitable for adequately addressing the specific research question at hand.
- Survival can be influenced by many factors, among which comorbidities or therapeutic strategies chosen to face the malignancy. No mention in this work is done about external factors that can influence survival so at this regard I can kindly suggest the analysis of this two works: https://pubmed.ncbi.nlm.nih.gov/32570240/ and https://pubmed.ncbi.nlm.nih.gov/37419854/
We appreciate the reviewer's contribution of these insightful articles. The references provided address the treatment of metastatic prostate cancer, a topic of utmost significance. However, it is essential to note that the applicability of these references to our population-based study is limited. Our study does not encompass data on specific pharmacological agents utilized for treatment, rendering the relevance of these articles somewhat diminished within the context of our research focus. As for the influence of comorbidities in our results, that is well covered in the discussion and limitations. Moreover, our choice of cause-specific survival instead of observed survival as the measured outcome diminishes the potential impact of comorbidities on survival.
Reviewer 3 Report
Comments and Suggestions for Authors
I have no further comments/remarks on the article.
Author Response
Thank you for reviewing our manuscript.